# Mood and Metabolic Health Status of Elderly Osteoporotic Patients in Korea: A Cross-Sectional Study of a Nationally Representative Sample

**DOI:** 10.3390/healthcare9010077

**Published:** 2021-01-14

**Authors:** Hyen Chul Jo, Gu-Hee Jung, Seong-Ho Ok, Ji Eun Park, Jong Chul Baek

**Affiliations:** 1Department of Obstetrics and Gynecology, Gyeongsang National University Changwon Hospital, Changwon-si 51472, Korea; 73hccho@gnuh.co.kr (H.C.J.); parkjieun@gnuh.co.kr (J.E.P.); 2College of Medicine, Gyeongsang National University, Jinju 52828, Korea; jyujin2001@gnu.ac.kr (G.-H.J.); mdoksh@gnu.ac.kr (S.-H.O.); 3Department of Orthopaedic Surgery, GNUCH, Changwon-si 51472, Korea; 4Medical ICT Convergence Research Center, Institute of Health Sciences, College of Medicine, GNU, 460 Iksan dea-ro, Iksan City, Jeonbuk 13557, Korea; 5Department of Anesthesiology and Pain Medicine, GNUCH, Changwon-si 51472, Korea

**Keywords:** elderly, osteoporosis, dyslipidemia, depression, hyperuricemia, abdominal obesity

## Abstract

This study aimed to investigate the association between osteoporosis and comorbidity, which are very common in Korea, and develop a treatment strategy to improve bone health based on the findings of the Korean National Health and Nutritional Examination Surveys (KNHANES). This study was based on data obtained from 4060 subjects (1755 males, 2305 females) aged above 60 years in the KNHANES (2016–2017). Well-trained medical staff performed the standard procedures and measured several variables including height, weight, and waist circumference. Interviews and laboratory tests were based on the diagnosis of hyperuricemia, dyslipidemia, type 2 diabetes mellitus (T2DM), osteoporosis, and depression. Comorbidities were defined as a self-reported physician diagnosis. The association of osteoporosis with depression and metabolic disease was assessed statistically using the complex sample analysis method of SPSS. The presence of osteoporosis, dyslipidemia, T2DM, hyperuricemia, obesity, abdominal obesity, and depression was 6.1 ± 0.5%, 15.2 ± 0.7%, 6.5 ± 0.4%, 13.4 ± 0.7%, 30.8 ± 0.8%, 19.4 ± 0.9%, 4.0 ± 0.2%, respectively. After adjusted by age, osteoporotic subjects were significance in the presence of abdominal obesity (*p* = 0.024, OR 0.80), hyperuricemia (*p* = 0.013, OR 0.68), dyslipidemia (*p* < 0.001, OR 1.84), and depression (*p* < 0.001, OR 2.56), respectively. Subgroup analyses showed dyslipidemia (female subjects, *p* < 0.001, OR 1.04; male subjects, *p* = 0.94, OR 1.09) and depression (female subjects, *p* < 0.001, OR 1.76; male subjects, *p* = 0.51, OR 0.62) were associated with osteoporotic female subjects but not in male subjects. The comorbidity of dyslipidemia and depression in female subjects was associated with osteoporosis and an odds ratio was 13.33 (95% CI: 8.58–20.71) (*p* < 0.001). The comorbidity of abdominal obesity (female subjects, *p* = 0.75, OR 0.97; male subjects, *p* = 0.94, OR 1.02) and hyperuricemia (female subjects, *p* = 0.27, OR 0.81; male subjects *p* = 0.07, OR 0.35) was not associated with osteoporosis in both Subgroup. The result of this study shows a strong dependency of comorbidity with dyslipidemia and depression in elderly women with osteoporosis. Therefore, efforts to improve dyslipidemia and depression might prevent compromised bone health.

## 1. Introduction

As some of the current world population ages, osteoporosis and osteoporosis-related fractures (ORF) pose a significant public health concern worldwide, including South Korea. Concerning the presence of osteoporosis, there are grossly two kinds of risk factors including fixed ones such as age or gender and modifiable ones including poor personal lifestyle choices and a medical conditions which would affect bone health. It is already known that at constant bone mineral density (BMD), the risk of fracture increases four-fold with a 20-year increase in age. [1,2] Therefore, reduced bone mass is obviously one of the many factors associated with the increased risk of fractures in old age. Recently, advances in knowledge including changes associated with age and body metabolism which could alter the bone structure, highlighted the importance of non-mass factors including bone quality and bone strength [2,3,4]. Thus, appropriate prevention and management of osteoporosis begins with awareness of individualized risk factors that compromise the status of bone health.

Recent evidence derived from animal [4,5] and human studies [5,6,7,8,9] suggests that obesity-derived metabolic alterations may also be risk factors for compromised bone strength by inducing inflammation and calcium excretion. However, studies report inconsistent results of the association between metabolic syndrome and osteoporosis [10]. Nonetheless, epidemiological studies [11,12,13,14] relating metabolic syndrome to osteoporosis have been few performed. Currently, South Korea is a rapidly aging society and bone health issues are an emerging and serious public health burden. Furthermore, in the aged society, there is an increase in the number of patients diagnosed with depression along with comorbid conditions that affect outcomes such as health-related quality of life, disability, and mortality. Recently, a cross-sectional study reported that diagnosed depression is associated with metabolic syndrome significantly higher in women than in men [15].

Given the substantial progress in the association between osteoporosis and chronic medical conditions, the management of patients with osteoporosis and the prevention of ORF might be combined with efforts to lower the risk factors compromising bone health. Therefore, we investigated the association of osteoporosis with comorbidity using a cross-sectional study of nationally representative sample populations (Korean National Health and Nutritional Examination Surveys; KNHANES) in an effort to develop appropriate treatment strategies for bone health improvement.

## 2. Materials and Methods

### 2.1. Study Design and Subjects

In this analysis, the data of third-year participants of KNHNES VI (2016) and first-year subjects of KNHANES VII (2017) were enrolled. Among the 16,277 targeted individuals, subjects who were under 60 years of age were excluded. The final 4060 participants (males, 1755; females, 2305) who underwent full health interview and survey were included.

Well-trained medical staff performed the standard procedures and measured several variables including height, weight, waist circumference, and body mass index (BMI; kg/m^2^), which was calculated using measured height and weight. Waist circumference (WC) was measured from the narrowest point between the borders of the rib cage and the iliac crest during minimal respiration. Based on the Asian-Pacific criteria of World Health Organization [16,17], the subjects were divided into normal (non-obese) (BMI < 25kg/m^2^) and obese (BMI ≥ 25.0 kg/m^2^) groups. The criterion of abdominal obesity, which was adopted by the Korean Society for the Study of Obesity, was followed: WC ≥ 90 cm for men, and WC ≥ 85 cm for women [18]. Laboratory parameters were measured using blood samples collected from the antecubital vein in the morning after an overnight fast. Total cholesterol, triglyceride (TG), low-density lipoprotein-cholesterol (LDL-direct), and urate levels of all the participants were measured enzymatically. To measure the glucose level, fasting plasma glucose and glycosylated hemoglobin (HbA1c) were used. Hyperuricemia was defined as a serum uric acid level greater than 7.0 mg/dl in men and greater than 6.0 mg/dl in women [19].

### 2.2. Ethical Approval

This cross-sectional study was approved by the Institutional Review Board of our institution and was based on a nationally representative cross-sectional and population-based surveys conducted by the Korea Centers for Disease Control and Prevention (KCDC).

### 2.3. Health Interview Questionnaire and Definition of Chronic Illness

The self-reported health interview questionnaire was designed to collect information related to medical conditions and other comorbidities using a face-to-face interview method by survey staff members who were required to complete an intensive training course. The presence of chronic illnesses including osteoporosis, depression, diabetes and dyslipidemia was defined as previous diagnosis by medical doctors. Dyslipidemia was defined as a LDL cholesterol level ≥130 mg/dL or HDL cholesterol level ≤40 mg/dL. T2DM was defined as HbA1c ≥6.5% or a fasting plasma glucose level ≥126 mg/dL. A physician’s diagnosis confirmed a patient’s disease status. “Have you ever been diagnosed with osteoporosis by a physician?,” answering “Yes” to the question were defined as osteoporotic patients. Depression included not only whether participants had been physician-diagnosed, but also whether they had experienced depression symptoms. answering “Yes” to the question were defined as experiencing depression symptoms.

### 2.4. Statistical Analysis

Using household registries, the KNHANES stratified households into sampling units using a multistage, probability-based sampling design based on sex, age, and geographic area [20]. To ensure an equal probability of selection and representation of the entire Korean population, statistical weights were assigned to each participant [20,21]. The study population was stratified into two groups depending on the presence of osteoporosis. Differences in demographic and anthropometric characteristics of the two groups were compared using a complex sample analysis method including the general linear model for quantitative variables, logistic regression model for qualitative variables, and others (SPSS Inc., version 25.0, Chicago, IL, USA). The estimated rates and mean values of demographic data and anthropometric measurement were obtained with standard errors. Complex sample design was used to identify factors associated with osteoporosis in patients with comorbid chronic illnesses including dyslipidemia, type 2 diabetes mellitus (T2DM), and depression. The odds ratio (OR) and 95% confidence interval (CI) were calculated with each characteristic adjusted to determine the associated risk of osteoporosis between groups. Two-sided *p*-values of <0.05 were considered significant.

## 3. Results

### 3.1. Overall Characteristics of Total Subjects with or without Osteoporosis

The mean age of the study population was 69.6 ± 0.1 years: 68.8 ± 0.2 years for male subjects, and 70.2 ± 0.19 years for female subjects. The mean height of all the subjects in the study population was 158.5 ± 0.56 cm: 166.1 ± 0.2 cm for males, and 152.5 ± 0.19 cm for females. The mean weight of all the subjects was 60.9 ± 0.4 kg: 65.8 ± 0.29 kg for males, and 56.9 ± 0.2 kg for females. The waist circumference of the subjects was 85.0 ± 0.21 cm: 86.5 ± 0.23 cm for males, and 83.8 ± 0.24 cm for females. The mean BMI was 24.2 ± 0.06: 23.8 ± 0.73 for males, and 24.5 ± 0.08 for females.

The presence of osteoporosis, dyslipidemia, T2DM, hyperuricemia, obesity, abdominal obesity, and depression was 6.1 ± 0.5%, 15.2 ± 0.7%, 6.5 ± 0.4%, 13.4 ± %0.7, 30.8 ± 0.8%, 19.4 ± 0.9%, 4.0 ± 0.2%, respectively. The total subjects were diagnosed with osteoporosis at an average age of 60.4 ± 0.62 years: 66.9 years ± 1.98 for males, and 63.66 years ± 0.59 for female subjects. T2DM was diagnosed in the mean age of 54.4 ± 0.59 for total subjects: 57.7 years ± 0.83 for male subjects, and 62.3 ± 0.85 for female subjects. The dyslipidemia was diagnosed in the mean age of 45.0 years ± 0.56 for total subjects: 60.9 years ± 0.98 for male subjects, and 62.1 years ± 0.49. The depression was diagnosed in the mean age of 35.9 years ± 1.86 for total subjects: 68.0 years ± 3.6 for male subjects, and 58.0 years ± 1.81 for female subjects.

Based on the presence of osteoporosis, the estimated means of comorbid disease and anthropometric measurements are presented in Table 1. In total subjects, all the presence rates of comorbid diseases including obesity, abdominal obesity, hyperuricemia, dyslipidemia, and depression were statistically significant difference between males and female subjects. Concerning the presence of osteoporosis, obesity, T2DM, and BMI were not significant between subjects with and without osteoporosis (Table 1).

### 3.2. Crosstabulation Tables for Osteoporosis Frequency by Comorbid Diseases

Using the Complex Samples Crosstabs procedure, a cross-tabulation of osteoporosis frequency based on comorbid diseases was performed. The odds ratios, and Chi-Square ratios (χ^2^) were computed. As shown in Table 2, the male subjects showed a statistically significant obesity and a negative dependency was observed (χ^2^ = 3.448, *p* = 0.039). In female subjects, the positively correlated values included dyslipidemia (χ^2^ = 11,347, *p* = 0.001) and depression (χ^2^ = 15.747, *p* < 0.001).

### 3.3. Logistic Regression Analysis of Comorbid Diseases with Osteoporosis

Univariable and multivariable logistic regression analyses were used to identify factors associated with osteoporosis, and the models were adjusted for age. The results of logistic regression analyses are summarized in Table 3. Based on univariable analysis adjusted of age in total subjects, osteoporosis was significantly associated with abdominal obesity (odds ratio [OR] 0.80; 95%CI, 0.65–0.97, *p* = 0.024), hyperuricemia (OR 0.68; 95% CI, 0.49–0.92, *p* = 0.013), dyslipidemia (OR 1.84; 95% CI 1.57–2.15, *p* < 0.001), and depression (OR 2.56; 95% CI, 1.89–3.47, *p* < 0.001) (Table 3). According to sex, the univariate analysis of male subjects was only significant with obesity (OR 0.53; 95% CI, 0.29–0.98, *p* = 0.041) and age adjusted OR was not significant with comorbidity (Table 4). Age adjusted OR was significant with dyslipidemia (OR 1.04; 95% CI, 1.03–1.06, *p* < 0.001) and depression (OR 1.76; 95% CI, 1.28–2.40, *p* < 0.001) in female subjects (Table 5). Thus, subjects with dyslipidemia and depression were stratified as a high-risk group in this study. The crosstabulation of osteoporosis frequency in high-risk subjects showed a positive dependency (χ^2^ = 218.011, *p* < 0.001, odds ratio = 13.330; 95% CI, 8.580–20.710) (Table 6). Abdominal obesity and hyperuricemia were negatively correlated with osteoporosis, albeit insignificantly (*p* = 0.173). A trend analysis using logistic regression analysis of osteoporotic female subjects with dyslipidemia and/or depression showed statistical significance (Table 7). When the subject was in high risk (with dyslipidemia and depression), the relative risk (OR) was 1.98 (95% CI, 1.35–2.90, *p* = 0.000) compared with non-risk subjects. In addition, if subjects had a risk factor of dyslipidemia or depression, the relative risk was 1.51 (95% CI, 1.27–1.80, *p* < 0.001) compared with non-risk subjects.

## 4. Discussion

Currently, due to rapid aging and an increase in obesity in certain populations, the management of related sociomedical issues is critical. Recent studies [4,5,6,7,8,9] suggest that obesity-derived metabolic alterations including dyslipidemia, glucose intolerance, and T2DM may also be risk factors for compromised bone health. Although Korea has entered the era of a super-aged and obese society, few studies [6,7,8,9,22] have been performed to investigate the population for age-related diseases. Therefore, in this cross-sectional study based on nationally representative sample populations, we established an independent relationship between osteoporosis and chronic diseases including general and abdominal obesity, dyslipidemia, T2DM, hyperuricemia and depression, reflecting the current scenario in Korea. Our results confirmed previous data correlating metabolic syndrome and osteoporosis, which suggested that (1) dyslipidemia was strongly associated with the presence of osteoporosis; and (2) general and abdominal obesity was the protector of BMD in both men and women. Notably, we found that depression was closely related to osteoporosis. If depression were comorbid with dyslipidemia in female subjects aged more than 60 years, the odds ratio of osteoporosis was 13.33 (95% CI, 8.58–20.71). Thus, the current study provided further evidence that treatment of patients with dyslipidemia and depression might improve bone health.

The association between metabolic syndrome and osteoporosis has been disputed. Recent evidence based on animal^4^ and human studies [6,9,23,24] has suggested that obesity-derived metabolic alterations may also be risk factors for compromised bone strength, although the findings were heterogeneous. Treatment with medications (statins) to lower blood cholesterol levels has been shown to inhibit osteoclastic bone resorption comparable to bisphosphonates [23,24] and activated osteoblasts by inducing the synthesis of bone morphogenetic protein-2 [25,26]. Similar results have also been reported in clinical studies showing beneficial effects on blood lipids, and in osteoporosis and fracture prevention [23,27,28]. In our study, although there was no consideration of whether or not to treat it, dyslipidemia was also negatively associated with osteoporosis (OR 1.84, *p* < 0.001). However, the level of LDL-direct was not statistically significant (*p* = 0.358). Alay et al. reported a negative dependency between osteoporosis and hyperlipidemia [29], which is similar to this study’s results. The association between osteoporosis and dyslipidemia reported in previous studies is heterogenous. Yamaguchi et al. reported a positive association between LDL and BMD levels [30], but Akram Ghadiri-Anari et al. reported no significant relationship [31]. Some evidence has illustrated an association of osteoporosis with dyslipidemia, but information regarding the impact of dyslipidemia is inconclusive yet inextricably linked to other potential factors, including healthy lifestyle, metalobic, nuturitional, and psychosocial parameters. The exact relationships between these factors is poorly understood. DM and osteoporosis were regarded as separate pathological processes, but it is recently thought that osteoporosis can develop as a complication of diabetes [32]. Diagnosed T2DM was consistently not correlated with the presence of osteoporosis in total subjects of this study (*p* = 0.981). Furthermore, the dependency was not statistically significant (*p* = 0.442), even though the analysis involved selected subpopulations, which were defined as female subjects aged more than 40 years. Unlike type 1 DM, which has decreased BMD, T2DM does not show a decrease in BMD [33]. Normal or elevated BMD in T2DM, risk of fracture is higher than that of nondiabetic patients. These facts implicate that the mechanism of fracture in T2DM is more complex biochemical activities and may be affected by multiple confounding mechanisms such as antidiabetic medications, associated renal pathology, genetics, and life style. Further investigations are needed on the possible covariate interactions and effects on the association between T2DM with fracture risk.

Hyperuricemia was independently associated with BMD and fractures, supporting a protective role for uric acid in bone metabolism disorders [34,35]. Proinflammatory cytokines up-regulate the receptor for nuclear factor-κB ligand, leading to increased bone resorption [36]. In our study, although the value of hs-CRP was not correlated with osteoporosis (*p* = 0.433), hyperuricemia was correlated inversely with osteoporosis (*p* = 0.013). However, interpretation is not definitive given the fact that the use of hyperuricemia medication was not considered. Interestingly, high serum uric acid levels were reported as a potential risk factor increasing the presence of metabolic syndrome and its components, including central obesity, hypertension, dyslipidemia, diabetes, and insulin resistance, which were inversely related with osteoporosis [37,38]. Thus, further studies are needed to determine the protective role of asymptomatic hyperuricemia in bone health among middle-aged and older adults.

Koreans also suffer from depression as life expectancy increases and the presence of depression has continuously increased. According to a report on the current status of the Korea National Health and Nutrition Examination Survey (KNHANES) [39], the presence of adult depression (in individuals aged over 19 years, standardized) in 2016 was 4.1% in males and 7.0% in females and the highest presence was among those in their 70s. The presence of hypercholesterolemia in adults aged 30 years or older was 19.9% and the awareness rate was just 58.4% [40]. As is well known, osteoporosis is also a silent disease before osteoporosis-related fractures occur. We assessed the association of osteoporosis with diagnosed depression and dyslipidemia as a risk factor in adults. Our study demonstrated that diagnosed depression and dyslipidemia were strongly correlated with osteoporosis and were risk factors in Korean women aged more than 60 years based on multivariate logistic regression analysis adjusted for age (OR 1.98; 95% CI 1.352–2.90, *p* = 0.000). Therefore, in addition to the well-known risk factors such as age and gender, chronic conditions such as dyslipidemia and depression were strong and silent risk factors in adults. Thus, in view of the complications associated with anti-resorptive medications (bisphosphonates), osteoporosis prevention and management should be included to lower these silent risk factors.

### Limitations of the Study

Although our study was based on nationally representative samples from Korea, there were several fundamental limitations. First, the annual sample size might be small to produce reliable estimates of detailed demographic or geographic information in the subgroups.^20^ Second, because KNHANES was designed as a cross-sectional study, a longitudinal follow-up of survey participants was limited. Third, excluding non-responders leads to biased estimates if they carried significantly different characteristics compared with responders. Forth, data used in this study were based on self-report of physician diagnosis without corroborating DEXA findings, so the possibility of false negatives cannot be completely verified. A follow-up study would be needed to confirm the accurate diagnoses using objective data and validate this study.

Nevertheless, our results represent valuable data for monitoring of the risk factors for osteoporosis and identification of target groups of warranting prompt interventions. Considering that depression, dyslipidemia, and osteoporosis were diagnosed at an average age of 47.6 ± 0.75 years, 54.51 ± 0.44 years, and 60.8 ± 0.42 years, respectively, in subjects aged above 30 years, the intervention points preventing from escalating poor bone health driven by the risk factors need to be addressed in a longitudinal follow-up study in future.

## 5. Conclusions

In conclusion, in our study osteoporosis had a strong correlation with dyslipidemia and depression in elderly individuals aged above 60 years. Considering the difference of diagnostic age of dyslipidemia and depression, these medical conditions should be actively managed to eliminate the compromising effect on bone health. In addition to antiresorptive medications to maintain bone mass, prophylactic interventions of lifestyle to lower the presence of dyslipidemia and depression might reduce the risk of osteoporosis. In elderly osteoporosis female patients, not only bone health, but also metabolic health management is essential.

## Figures and Tables

**Table 1 healthcare-09-00077-t001:** Overall characteristics and comparison of total subjects, male subjects and female subjects with or without osteoporosis.

Characteristics	Total (*n* = 4060)		*p* Value ^3^	Male (*n* = 1755, Mean ± SD			Female (*n* = 2305, Mean ± SD)		*p* Value ^5^
	N-OS ^1^ (*n* = 3199)	OS ^2^ (*n* = 861)		N-OS (*n* = 1702)	OS (*n* = 53)	*p* Value ^4^	N-OS (*n* = 1497)	OS (*n* = 808)	
Age	69.1 ± 0.1	71.3 ± 0.3	0.000	68.8 ± 0.2	71.1 ± 0.9	0.138	69.5 ± 0.3	71.3 ± 0.3	0.000
Height, cm	160.0 ± 0.6	152..7 ± 0.3	0.000	166.2 ± 0.2	164.3 ± 1.3	0.651	152.8 ± 0.2	151.9 ± 0.3	0.586
Weight, kg	62.19 ± 0.4	56.3 ± 0.4	0.000	66.0 ± 0.3	62.1 ± 1.1	0.817	57.6 ± 0.3	55.9 ± 0.3	0.435
BMI (kg/m^2^)	24.2 ± 0.7	24.1 ± 0.1	0.575	23.8 ± 0.7	23.0 ± 0.3	0.639	24.6 ± 0.1	24.2 ± 0.1	0.777
Obesity (BMI ≥ 25) (%)	38.5 ± 1.0	37.4 ± 2.0	0.656	34.9 ± 1.3	22.2 ± 5.6	0.043	42.7 ± 1.6	38.5 ± 2.1	0.045
Waist circumference (cm)	85.4 ± 0.2	83.6 ± 0.4	0.000	86.5 ± 0.2	84.8 ± 1.3	0.426	84.0 ± 0.3	83.6 ± 0.4	0.081
Abdominal obesity (%)	28.9 ± 0.9	24.7 ± 1.8	0.034	33.4 ± 1.3	32.8 ± 6.6	0.139	23.7 ± 1.2	24.1 ± 1.9	0.183
Hyperuricemia (%)	11.9 ± 0.7	9.0 ± 1.1	0.024	13.4 ± 1.0	5.2 ± 2.9	0.073	10.1 ± 0.9	9.3 ± 1.2	0.668
Dyslipidemia (%)	30.3 ± 1.2	42.6 ± 2.2	0.000	25.0 ± 1.3	25.5 ± 6.8	0.907	36.5 ± 1.6	43.8 ± 2.3	0.005
Type 2 DM ^6^ (%)	20.1 ± 1.0	20.2 ± 1.6	0.981	20.1 ± 1.4	20.0 ± 6.1	0.556	19.9 ± 1.4	20.2 ± 1.6	0.842
Depression (%)	4.6 ± 0.4	11.4 ± 1.1	0.000	2.6 ± 0.4	1.6 ± 1.2	0.577	7.1 ± 0.6	12.0 ± 1.2	0.001

^1^ Non-Osteoporosis subjects; ^2^ Osteoporosis subjects; ^3^ Comparison between non-osteoporosis and osteoporosis total subjects; ^4^ Comparison between non-osteoporosis male subjects and osteoporosis male subjects. ^5^ Comparison between non-osteoporosis female subjects and osteoporosis female subjects.^6^ Diabetes Mellitus.

**Table 2 healthcare-09-00077-t002:** Crosstabulation tables for osteoporosis frequency by comorbid diseases.

**Cormorbid Diseases**		**Male (Unweighted Count)**		***X*^2^ (*p* Value)**	**Female (Unweighted Count)**		***X*^2^** **(*p* Value)**
		**N-OS**	**OS**		**N-OS**	**OS**	
Obesity	Undiagnosed	1116	40	3.448 (0.039)	846	503	3.801 (0.123)
	Diagnosed	584	13		650	300	
Abdominal obesity	Undiagnosed	1276	37	0.007 (0.934)	1128	617	0.04 (0.858)
	Diagnosed	570	16		365	188	
Hyperuricemia	Undiagnosed	1406	45	2.666 (0.070)	1248	690	0.367 (0.590)
	Diagnosed	226	4		146	62	
Dyslipidemia	Undiagnosed	1297	42	0.009 (0.929)	941	458	11.347 (0.001)
	Diagnosed	405	11		556	350	
Type 2 DM	Undiagnosed	1333	42	0.003 (0.955)	1195	646	0.025 (0.881)
	Diagnosed	369	11		302	162	
Depression	Undiagnosed	1655	51	0.165 (0.539)	1384	710	15.747 (0.000)
	Diagnosed	47	2		113	97	

**Table 3 healthcare-09-00077-t003:** Logistic regression analysis of comorbid diseases with osteoporosis adjusted by age in total subjects.

Cormorbid Diseases	Univariate OR (95% CI)	*p* Value	Age Adjusted OR (95% CI)	*p* Value
Obesity (BMI ≥ 25) (%)	0.96(0.79–1.16)	0.656	0.99 (0.81–1.20)	0.880
Abdominal obesity (%)	0.80(0.66–0.98)	0.034	0.80 (0.65–0.97)	0.024
Hyperuricemia (%)	0.73(0.56–0.96)	0.024	0.68 (0.49–0.92)	0.013
Dyslipidemia (%)	1.71(1.47–1.99)	<0.001	1.84 (1.57–2.15)	<0.001
Type 2 DM (%)	1.00(0.82–1.23)	0.981	0.92 (0.78–1.13)	0.442
Depression (%)	2.63(1.96–3.54)	<0.001	2.56 (1.89–3.47)	<0.001

OR: odds ratio, CI: confidence interval.

**Table 4 healthcare-09-00077-t004:** Logistic regression analysis of comorbid diseases with osteoporosis adjusted by age in male subjects.

Cormorbid Diseases	Univariate OR (95% CI)	*p* Value	Age Adjusted OR (95% CI)	*p* Value
Obesity (BMI ≥ 25) (%)	0.53(0.29–0.98)	0.041	0.59 (0.32–1.09)	0.091
Abdominal obesity (%)	0.98(0.54–1.78)	0.934	1.02 (0.56–1.86)	0.942
Hyperuricemia (%)	0.36(0.11–1.14)	0.082	0.35 (0.11–1.10)	0.073
Dyslipidemia (%)	1.03(0.52–2.05)	0.929	1.09 (0.56–2.15)	0.946
Type 2 DM (%)	0.98(0.47–2.04)	0.955	0.98 (0.47–2.03)	0.795
Depression (%)	0.64(0.15–2.72)	0.542	0.62 (0.15–2.58)	0.512

**Table 5 healthcare-09-00077-t005:** Logistic regression analysis of comorbid diseases with osteoporosis adjusted by age in female subjects.

Cormorbid Diseases	Univariate OR (95% CI)	*p* Value	Age Adjusted OR (95% CI)	*p* Value
Obesity (BMI ≥ 25) (%)	0.84(0.67–1.05)	0.122	0.83 (0.67–1.04)	0.105
Abdominal obesity (%)	1.02(0.81–1.28)	0.858	0.97 (0.77–1.21)	0.753
Hyperuricemia (%)	0.91(0.65–1.28)	0.590	0.81 (0.56–1.18)	0.274
Dyslipidemia (%)	1.35(1.14–1.61)	0.001	1.04 (1.03–1.06)	<0.001
Type 2 DM (%)	1.02(0.81–1.28)	0.881	0.91 (0.72–1.14)	0.398
Depression (%)	1.79(1.31–2.46)	<0.001	1.76 (1.28–2.40)	<0.001

**Table 6 healthcare-09-00077-t006:** Crosstabulation tables of high-risk female subjects ^1^ for osteoporosis frequency by comorbid diseases.

	**High Risk Group ^1^ Frequency** **(Mean ± SD)**	**General Group ^2^ Frequency (Mean ± SD)**	***X*^2^** **(*p* Value^2^)**	**Odds Ratio** **(95% CI)**
Non-osteoporosis	80	3119	218.011 (0.000)	13.330 (8.580–20.710)
Osteoporosis	56	804		

^1^ Female subjects with dyslipidemia and depression; ^2^ Female subjects without dyslipidemia and depression.

**Table 7 healthcare-09-00077-t007:** Logistic regression analysis of osteoporotic subjects with dyslipidemia and/or depression adjusted by age.

	Age Adjusted OR (95% CI)	*p* Value for Trend
Without dyslipidemia and depression	1.0	<0.001
With dyslipidemia or depression	1.51(1.27–1.80)	
With dyslipidemia and depression	1.98(1.35–2.90)	

## Data Availability

The data evaluated in this study cannot be uploaded publicly due to legal restrictions, concerns for patient privacy, and third-party ownership of the data by the Korea National Health & Nutrition Examination Survey (KNHANES). However, the data are directly obtainable upon request, either by accessing https://knhanes.cdc.go.kr/knhanes/eng/index.do or by emailing knhanes@korea.kr.

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
