# Peer review of "Mood and Metabolic Health Status of Elderly Osteoporotic Patients in Korea: A Cross-Sectional Study of a Nationally Representative Sample"

_healthcare, 2021, doi:10.3390/healthcare9010077_

Round 1

Reviewer 1 Report

This is a well-written paper, almost ready for publication. The author also provided an appropriate literature review and clear research design, as well as reliable conclusions, the reviewer only makes the following suggestions:
Lines 207-216, line 233-234: The paper points out the negative correlation between osteoporosis and dyslipidemia, but there is no correlation between type II diabetes and different age groups of women, which means there is no statistical correlation with osteoporosis. And these two findings  the authors suggest due to the complicated reaction of medication.
These sections the reviewer suggests the authors could add some more critical discussion, need more cross references in order to provide solid evidence, especially to add healthy life style or psychosocial additional considerations.

Author Response

Response to Reviewer 1 Comments

Point 1: Lines 207-216, line 233-234: The paper points out the negative correlation between osteoporosis and dyslipidemia, but there is no correlation between type II diabetes and different age groups of women, which means there is no statistical correlation with osteoporosis.

And these two findings the authors suggest due to the complicated reaction of medication.

These sections the reviewer suggests the authors could add some more critical discussion, need more cross references in order to provide solid evidence, especially to add healthy lifestyle or psychosocial additional considerations.

Response 1:

  • We appreciate the Reviewer’s insightful comments.
  • For the sake of more critical discussion and relevance of the main topic, we added the following sentence to revised manuscript Line 228-246

Alay et al. reported negative correlation between osteoporosis and hyperlipidemia, which are similar to this study result. But the association between osteoporosis and dyslipidemia reported in studies is heterogenous.  Yamaguchi et al. reported a positive association between LDL and BMD levels. But Akram Ghadiri-Anari et al. reported no significant relationship. Some evidence has illustrated association of osteoporosis with dyslipidemia, but information regarding the impact of dyslipidemia is inconclusive yet inextricably linked to other potential factors, including healthy lifestyle, metalobic, nuturitional and psychosocial parameters. The exact relationships between these factors is poorly understood. DM and osteoporosis were regarded as separate pathological processes, but it is recently thought that osteoporosis can develop as a complication of diabetes. Diagnosed T2DM was consistently not correlated with the prevalence of osteoporosis in total subjects of this study (p=0.981). Furthermore, the correlation was not statistically significant (p = 0.442), even though the analysis involved selected subpopulations, which were defined as female subjects aged more than 40 years. Unlike type 1 DM, which has decreased BMD, T2DM does not show a decrease in BMD. Normal or elevated BMD in T2DM, risk of fracture is higher than that of nondiabetic patients. These facts implicate that mechanism of fracture in T2DM is more complex biochemical activities and may be affected by multiple confounding mechanisms such as antidiabetic medications, associated renal pathology, genetics, and life style. Further investigations are needed  on the possible covariates interactions and effects on the association between T2DM with fracture risk.

Reviewer 2 Report

The study aims to explore the association between osteoporosis and different comorbidities in the Korean population through data obtained from the Korean National Health and Nutritional Examination Surveys. Before we can consider the manuscript for publication, there are several issues that need to be clarified by the authors.

Abstract

The numerous results that are written following this structure are not understood (p<0.001, OR 30 1.04; p=0.94, OR 1.09). If the authors are referring to the odd ratio of e.g. dyslipidemia, what do they mean with the second OR? This is the same for various results in this section.

Applicable for this section but also for the rest of the manuscript, and concerning the interpretation that is made of the result, I disagree with the authors in the sense that a Chi-square analysis reflects correlations between variables. What the Chi-square analysis shows us is whether there is dependency or independence between the variables included in the analysis, not whether they are correlated, and much less the direction in which this dependency is produced.

Methods

One of the major weaknesses of this study, and not covered in the article, is the method of identifying patients with osteoporosis and other comorbidities. This work only makes sense and has merit if the diagnosis of osteporosis is objective and accurate. Using a personal interview for this I believe carries enormous biases and makes the interpretation of the results very complicated. Were the authors able to contrast the diagnosis of osteoporosis in the histories of the patients by verifying the T-score?It is necessary to detail more concisely and clearly how the diagnosis of osteoporosis was obtained and if this was contrasted. Were participants being treated for their osteoporosis? Since when? 

This information must be taken into account in a work of this type and especially when generating logistic regression models trying to control for possible covariates.

The population was not stratified according to the prevalence of osteoporosis, the population was stratified according to the presence and/or absence of osteoporosis. This concept should be corrected throughout the manuscript.

Results. 

Line 130. “Based on the prevalence”; I insist that I do not agree with this approach. For me it would be "based on the presence of osteoporosis"

Table 1:In superscript 4, (comparison between males and females) What is being compared? What statistical tests have been used? (Because in the materials and methods section no appropriate test is indicated for quantitative variables and comparisons between groups)

Table 2 and 6 must be redone by indicating the raw frequencies (not the percentages).

At the bottom of table 6 there is a superscript 1 that is not included in the table

Author Response

We appreciate the Reviewer’s insightful comments.

Response to Reviewer 2 Comments

Point 1:  Abstract

The numerous results that are written following this structure are not understood (p<0.001, OR 30 1.04; p=0.94, OR 1.09). If the authors are referring to the odd ratio of e.g. dyslipidemia, what do they mean with the second OR? This is the same for various results in this section.

Response 1:  The former (p<0.001 , OR  1.04) is descriptive in female subjects and the latter ( p=0.94, OR 1.09) is descriptive in male subjects.

To clarify the meaning, we changed  sentence as follows

(female subjects, p<0.001, OR 1.04; male subjects, p=0.94, OR 1.09)

Point 2:

Applicable for this section but also for the rest of the manuscript, and concerning the interpretation that is made of the result, I disagree with the authors in the sense that a Chi-square analysis reflects correlations between variables. What the Chi-square analysis shows us is whether there is dependency or independence between the variables included in the analysis, not whether they are correlated, and much less the direction in which this dependency is produced.

Response 2:

  • We appreciate the Reviewer’s insightful comments.
  • As your comments was appropriate, we replaced correlations with dependency throughout the manuscript

Point 3:

Methods

One of the major weaknesses of this study, and not covered in the article, is the method of identifying patients with osteoporosis and other comorbidities. This work only makes sense and has merit if the diagnosis of osteporosis is objective and accurate. Using a personal interview for this I believe carries enormous biases and makes the interpretation of the results very complicated. Were the authors able to contrast the diagnosis of osteoporosis in the histories of the patients by verifying the T-score? It is necessary to detail more concisely and clearly how the diagnosis of osteoporosis was obtained and if this was contrasted. Were participants being treated for their osteoporosis? Since when? This information must be taken into account in a work of this type and especially when generating logistic regression models trying to control for possible covariates.

Response 3:

- thank you for your comments. KNHANES is a continuous survey with nationally representative samples of Korea, and the health interview, physical examination and nutrition survey are combined to assess associations between variables. KNHANES data are valuable sources for monitoring the changes in risk factors and diseases and identifying target groups in need of interventions. By the rule of KNHANES survey, the presence of osteoporosis and other chronic illness is assumed to be when diagnosed by a medical doctor.

  • We added the following sentence to revised manuscript Line 100-106

Dyslipidemia was defined as a LDL cholesterol level ≥ 130 mg/dL or HDL cholesterol level ≤40 mg/dL . T2DM was defined as HbA1c ≥  6.5% or a fasting plasma glucose level ≥  126 mg/dL. A physician’s diagnosis confirmed a patient’s disease status. “Have you ever been diagnosed with osteoporosis by a physician?,” answering “Yes” to the question were defined as osteoporotic patients. Depression included not only whether participants had been physician-diagnosed, but also whether they had experienced depression symptoms. answering “Yes” to the question were defined as experiencing depression symptoms.

As you pointed out, our date did not directly measure DEXA to diagnosis osteoporosis. According to the World Health Organization, BMD measured by DEXA is the standard in defining osteoporosis. We cannot contrast the diagnosis of osteoporosis in the histories of the patients by verifying the T-score. This is the limitation of our study. But most residents of the Republic of Korea (more than 98%) are covered by the National Health Insurance Service. DEXA was offered free of charge to the general Korean population (over 54 years old) enrolled in National Health Insurance program and covered by health insurance service. National health examination is mandatory in Korea. So KNHANES date representative of the presence of osteoporosis, although diagnosis made by the health interview questionnaire.

  • We added the following sentence to revised manuscript Line 278-281.

Data used in this study were based on self-report of physician diagnosis without corroborating DEXA findings, so the possibility of false negative cannot be completely verified. A follow-up study would need to confirm the accurate diagnoses using objective data and validate this study.

The population was not stratified according to the prevalence of osteoporosis, the population was stratified according to the presence and/or absence of osteoporosis. This concept should be corrected throughout the manuscript.

  • As your comments was appropriate, we replaced prevalence with presence throughout the manuscript

Point 4 : Results. 

Line 130. “Based on the prevalence”; I insist that I do not agree with this approach. For me it would be "based on the presence of osteoporosis"

- thank you for your kindly review, we revised the word as your recommended

We replaced prevalence with presence throughout the manuscript

Table 1:In superscript 4, (comparison between males and females) What is being compared? What statistical tests have been used? (Because in the materials and methods section no appropriate test is indicated for quantitative variables and comparisons between groups)

Thank you for your careful comments, we added the statistical method in material and method

- Table 1 was revised as the total subjects, male subjects, and female subjects

and statistically analyzed by general linear model and logistic regression model.

Table 2 and 6 must be redone by indicating the raw frequencies (not the percentages).

At the bottom of table 6 there is a superscript 1 that is not included in the table

: thank you for your comment, we reran the statistical analysis and described the raw frequency in Table 2 and 6

: and, insert the superscript 1 in the table 6